# Changes in lymph node surgery in breast cancer and preoperative drug prescription analysis for postoperative pain management: A retrospective, cross-sectional study

Eun Sol Kang[1], Yu-Cheol Lim[2], Bo-Hyoung Jang[3], Yoon Jae Lee[2], In-Hyuk Ha[2]*, Ye-Seul Lee[2]*

1 Jaseng Hospital of Korean Medicine, Gangnam-daero, Gangnam-gu, Seoul, Republic of Korea, 2 Jaseng Spine and Joint Research Institute, Jaseng Medical Foundation, Gangnam-daero, Gangnam-gu, Seoul, Republic of Korea, 3 Department of Preventive Medicine, College of Korean Medicine, Kyung Hee University, Kyungheedae-ro, Dongdaemun-gu, Seoul, Republic of Korea

* hanihata@gmail.com (IHH); yeseul.j.lee@gmail.com (YSL)

**Data Availability Statement:** The datasets generated during and/or analyzed during the current study are available in the HIRA-NPS

## Abstract

This study aimed to investigate the changes in lymph node surgery types and prescription patterns of postoperative medications for pain management in patients with breast cancer using national health insurance claim data from South Korea. The study population comprised patients with at least one record of a principal diagnosis of breast cancer (ICD-10 code: C50) from the national health insurance claim database between 2010 and 2019. Patients who underwent mastectomy or lumpectomy only once were selected for the analysis. Patients who underwent axillary lymph node dissection (ALND) with mastectomy or lumpectomy on the day of surgery were included in the ALND group, whereas those who underwent sentinel lymph node biopsy (SLNB) were included in the SLNB group. Prescription records of opioids before, after and on the date of breast cancer surgery were collected and categorized according to the opioid type. Multivariate logistic regression modeling was used to compare postoperative opioid prescriptions. The proportion of those undergoing ALND among 3,080 patients decreased consistently after 2014, while the proportion undergoing SLNB increased. Although the rate of pain medication prescription on the day of surgery was similar between the two groups, the rate of prescription of postoperative pain medication and anticancer agents was lower in the SLNB group than in the ALND group. Logistic regression modeling showed that the SLNB group had lower odds of receiving opioids than did the ALND group (Odds ratio (OR) = 0.727, Confidence Interval (CI) = 0.546–0.970). A consistent trend was observed when the model was adjusted for neoadjuvant chemotherapy and the use of preoperative pain medications (OR = 0.718, CI = 0.538–0.959). To manage postoperative pain and prevent chronic pain with minimal side effects, sufficient discussion among clinicians, patients, and other healthcare professionals is imperative, along with adequate treatment planning.

repository [http://opendata.hira.or.kr] and upon payment of a data request additional fee. Other researchers would be able to access these data in the same manner as the authors, i.e. once they provide a research proposal to HIRA with an official review or exemption letter from their IRB. The authors did not have any special access privileges that others would not have.

**Funding:** The author(s) received no specific funding for this work.

**Competing interests:** The authors have declared that no competing interests exist.

## Introduction

Breast cancer is a major cancer in terms of global incidence and mortality. According to the Global Cancer Statistics, 2020 estimates of cancer incidence and mortality, female breast cancer is the most commonly diagnosed cancer, accounting for 11.7% of new cancer cases [1]. The 2018 Korea Central Cancer Registry has reported that breast cancer is the fifth most common cancer in Korea among all cancer types (23,647, 9.7%) and is the most common malignancy among women, accounting for 20.5% of all cancer cases [2].

Breast cancer treatment regimens commonly include chemotherapy, hormone therapy, radiation therapy, and surgical resection, of which surgical management of breast cancer is the foremost option [3]. In the surgical management of breast cancer, axillary lymph node dissection (ALND) traditionally served as a pathological staging and therapeutic tool for patients with breast cancer, playing an essential role in the locoregional management of breast cancer. However, ALND is increasingly associated with complications, such as pain, cutaneous necrosis, collection of lymph fluid, wound infections, numbness or paresthesia, lymphedema, impairment of arm mobility, and shoulder stiffness [4]. In particular, postoperative pain is a major problem affecting the quality of life of patients with breast cancer [5]. According to previous studies, such as those by Schrenk et al. [6] and Peintinger et al. [7], patients who underwent sentinel lymph node biopsy (SLNB) experienced significantly lower postoperative pain than did those who underwent ALND. With the recent development of surgical and diagnostic techniques, SLNB has evolved into a highly useful tool for reducing negative postoperative outcomes and is the preferred choice, where applicable [8]. In addition, several clinical studies have reported that compared to other surgical methods, SLNB allows for the use of a smaller incision, but the oncological outcomes of the technique are not compromised [9–11].

Opioids, which are narcotic analgesics, are an irreplaceable component of pharmacotherapy for pain management; however, because they are highly addictive, their use has become one of the most widely studied national public health problems in the United States over the last 10 years [12, 13]. The use of opioids in South Korea is relatively lower compared with that in other Organization for Economic Co-operation and Development countries [14], and there have been no issues with mortality related to opioid prescriptions in the country to date [15]. However, opioids can be readily prescribed through outpatient visits to primary care physicians and secondary and tertiary hospitals in South Korea [16], and one study using a sample cohort in Korea showed a continuously increasing trend of opioid use from 2002 to 2015 [16]. Accordingly, the awareness of the potential hazards of opioid use is increasing in South Korea.

Most patients with breast cancer undergoing surgery as the mainstay of treatment are routinely discharged with opioid prescriptions for pain management, indicating a high probability of opioid exposure [17]. Previous studies have reported that surgical intervention is associated with the first exposure to opioids in several cases [18] and that the type of surgery for patients with breast cancer is associated with postoperative opioid analgesic prescriptions [19]. Moo et al. [17] reported that most patients undergoing lumpectomy/SLNB could be discharged after surgery with nonsteroidal anti-inflammatory drug/acetaminophen prescriptions alone without resorting to opioid prescriptions for pain control, thereby reducing the number of opioid prescriptions.

To date, there has been a study on differences in narcotic prescriptions as postoperative pain medication with types of surgery, such as mastectomy/lumpectomy [20], but there have been few studies examining differences in the prescription of postoperative pain medications according to the methods of lymph node surgery for patients with breast cancer. Several studies have reported significantly lower postoperative pain in patients who underwent SLNB than

in those who underwent ALND [6, 7]; however, whether this is related to the prescription of postoperative pain medications remains unclear.

To address these issues, this study investigated a 10-year trend (2010–2019) in surgical techniques applied for lymph node surgery among patients with breast cancer, focusing on postoperative pain management, and examined whether there was a difference in the prescription of pain medications according to the type of lymph node surgery.

## Materials and methods

### Data collection

In this study, data from January 2010 to December 2019 obtained from the Health Insurance Review and Assessment Service (HIRA)-National Patient Sample (NPS) (provided by the HIRA of South Korea) were used. The HIRA data are insurance claim data generated when a healthcare service provider requests reimbursement from the national health insurance (NHI) for the healthcare service provided. The data provide detailed and diverse information, such as details of the healthcare service provided (treatment, procedure, tests and examinations, and prescription drugs), diagnosis, the amount paid by the payer, patient's out-of-pocket cost, patient population characteristics, and information on healthcare service providers.

The HIRA-NPS data extraction process is based on a stratified randomized sampling method with stratified age and sex groups of the sampled population, which is a 2% sample (approximately 1,000,000 persons) randomly extracted from the total national population of South Korea (those enrolled in the NHI or Medical Aid). The data are secondary data statistically sampled from raw data, with all personal information and information of the associated corporate bodies being de-identified. It is constructed from information on healthcare services claimed over 1 year from the date of starting medical care in each corresponding year [21, 22].

### Study design and population

Patients who had NHI claims issued at least once with a malignant neoplasm of the breast (ICD-10 code: C50) as the principal diagnosis over the 10-year period from 2010 to 2019 and those who underwent mastectomy or lumpectomy once between February 1 and November 30 in the corresponding year (index period) were included in the analysis. To exclude patients who underwent two or more surgeries in the corresponding year, those who underwent mastectomy, lumpectomy, or lymph node surgery on a date other than the surgery date were excluded from the study patient population.

The preoperative period was defined as 1 month prior to the surgery, excluding the day of surgery; the index date was defined as the day of surgery; and the postoperative period was defined as 1 month after surgery. Based on these definitions, the medication prescription history of the patients for the preoperative period, index date, and postoperative period was investigated (Fig 1). Patients who underwent ALND along with mastectomy or lumpectomy on the day of surgery were included in the ALND group, whereas those who underwent SLNB along with breast cancer surgery were included in the SLNB group.

### Study outcomes

The baseline characteristics of the patients included age, sex, payer type, type of surgery, and type of preoperative pain medication. Age was categorized into six groups: <30, 30–39, 40–49, 50–59, 60–69, and ≥70 years. Payer type was categorized as NHI or Medicaid, and preoperative pain medication was categorized into four types. The number of patients and percentage of patients based on the type of surgery were estimated for each year during the study period.

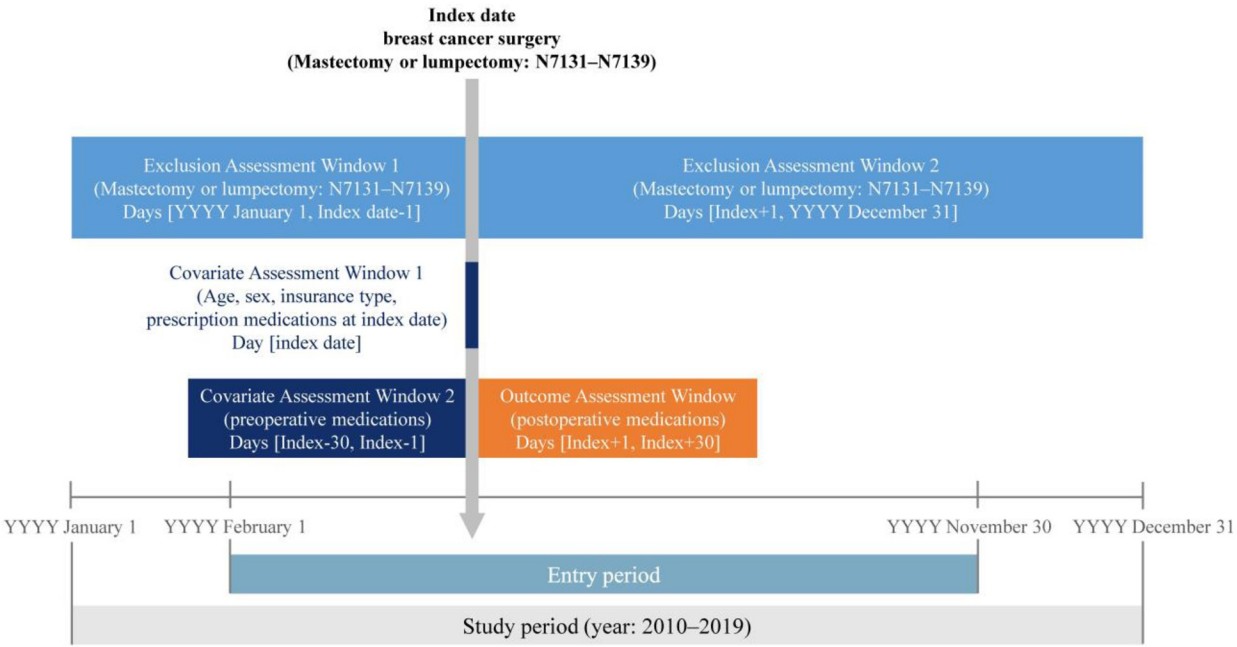

**Fig 1. Study design.**

The pre- and postoperative period medications (medications prescribed during 1 month before and after breast cancer surgery and dispensed from the pharmacy and hospital) and the medications on the index date (day of surgery) were classified into nine categories according to the Anatomical Therapeutic Chemical (ATC) code of the active ingredients of each medication. The frequency of prescriptions for each category was analyzed from 2010 to 2019. Among these, pain medications were further subdivided according to the World Health Organization (WHO) three-step analgesic ladder: non-opioid analgesics for the first step, tramadol for the second step, and opioids for the third step [23]. The pain medications were assigned into categories based on the drug that provided the strongest pain relief among the pain medications prescribed during each period. S1 Table lists the ATC code classifications.

Another outcome was the number of opioid prescriptions during the postoperative period according to the type of surgery. Opioids included morphine, hydromorphone, oxycodone, dihydrocodeine, codeine, hydrocodone, pethidine, fentanyl, pentazocine, buprenorphine, butorphanol, nalbuphine, tapentadol, sufentanil, remifentanil, and tramadol. Although tramadol is generally included in the opioid category, it is not classified as a narcotic analgesic in South Korea; thus, tramadol was separately analyzed.

## Statistical analyses

The number of patients and percentages (%) under the categorical variables of baseline characteristics, ALND and SLNB (type of surgery) cases by year, and medications for different prescription periods were reported. Multivariate logistic regression analysis was performed to calculate the odds ratios (ORs) for opioid prescriptions and 95% confidence intervals (95% CIs). The year of surgery, age, payer type, preoperative period of pain medications for each step based on the WHO analgesic ladder, and status of anticancer agent utilization in the preoperative period were considered covariates in the analysis [24]. Model 1 was adjusted for the

year of surgery, and Model 2 was adjusted for age, payer type, and year of surgery. Model 3 was adjusted for preoperative pain medications and preoperative anticancer drugs in addition to the covariates of Model 2. All analyses were performed using the SAS software (version 9.4, SAS Institute, Cary, NC, USA), and a two-sided p-value of <0.05 was considered statistically significant.

## Ethical considerations

This study was exempt from institutional review board (IRB) review by our hospital's IRB (IRB file no. JASENG 2022-10-008) because we used existing publicly available data, and the patients' information could not be directly identified through identifiers linked to the patients. Patient consent was not required, as this study used a secondary database with de-identified patient information.

## Results

A total of 29,896 patients with NHI claim data with malignant neoplasm of the breast (ICD-10 code: C50) as the principal diagnosis at least once during the 10-year period from 2010 to 2019 were identified. Among these patients, those who did not undergo mastectomy or lumpectomy from February 1 to November 30 for each applicable year (n = 26,729) and those who underwent mastectomy, lumpectomy, or lymph node surgery on a day other than the index date (n = 87) were excluded from the analysis. A total of 3,080 patients were included in the final study sample (Fig 2).

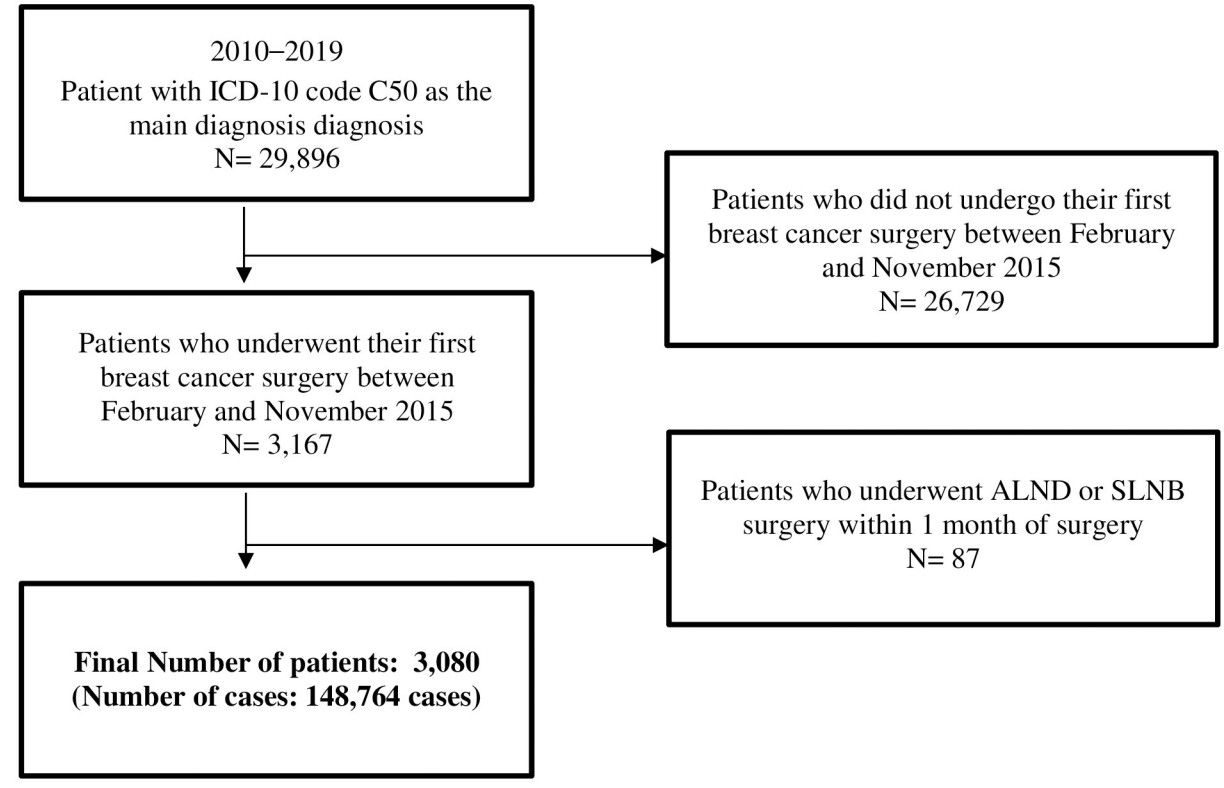

**Fig 2. Study patient selection.**

Regarding the age group of patients with breast cancer included in this study, patients in their 40s accounted for the highest proportion (n = 1,118, 36.30%), followed by those in their 50s (n = 710, 23.05%) and 30s (n = 673, 21.85%). Among the age groups, patients in their 40s accounted for the highest proportion in both the ALND (n = 672, 37.19%) and SLNB (n = 446, 35.04%) groups, showing a similar distribution to that of the total patient sample. Most patients were women (female, n = 3,068 [99.61%]; male, n = 12 [0.39%]). The NHI was the dominant payer type (n = 2,961, 96.14%), and a small number of patients received Medicaid (n = 119, 3.86%). There were no significant differences between the ALND and SLNB patient groups in terms of age, sex, and type of insurance. The analysis of the prescription records during the preoperative period (1 month prior to the index date) for all breast cancer patients included in this study showed that the majority of patients (n = 2,745, 89.12%) were not prescribed any painkillers. Among those who received prescriptions, non-opioid analgesics were prescribed to 214 patients (6.95%), tramadol to 55 patients (1.79%), and opioids other than tramadol to 66 patients (2.14%). The percentage of patients prescribed preoperative non-opioid analgesics or tramadol was slightly higher in the ALND group than in the SLNB group (non-opioid analgesics: 8.30% vs. 5.03%, tramadol: 2.10% vs. 1.34%) (Table 1).

The examination of the annual trend of changes in the number of patients who underwent lymph node surgery revealed that the total number of patients (irrespective of the type of surgery) increased from 261 in 2010 to 359 in 2019, and the proportion of patients who underwent SLNB increased from 19.5% (n = 51) in 2010 to 77.7% (n = 279) in 2019. In contrast, the number of patients who underwent ALND decreased from 80.5% (n = 210) in 2010 to 22.3% (n = 80) in 2019 (Fig 3).

The prescription rates of pain-related medications and anticancer drugs were analyzed for the ALND/SLNB patient groups by categorizing them into the preoperative period, index date, and postoperative period (Table 2). The numbers of patients prescribed non-opioid analgesics during the postoperative period (for 1 month after surgery) were 362 (20.03%) in the ALND group and 222 (17.44%) in the SLNB group. Tramadol was prescribed during the postoperative period in 122 (6.75%) and 70 (5.50%) patients in the ALND and SLNB groups, respectively. The numbers of patients prescribed opioids were 91 (5.04%) in the ALND group and 48

**Table 1. Baseline characteristics of the study population.**

| Category | | Total (n = 3,080) | | ALND (n = 1,807) | | SLNB (n = 1,273) | |
|---|---|---|---|---|---|---|---|
| | | N | Percent | N | Percent | N | Percent |
| Age | <30 | 34 | 1.1 | 19 | 1.1 | 15 | 1.2 |
| | 30–39 | 334 | 10.8 | 220 | 12.2 | 114 | 9.0 |
| | 40–49 | 1,045 | 33.9 | 616 | 34.1 | 429 | 33.7 |
| | 50–59 | 906 | 29.4 | 540 | 29.9 | 366 | 28.8 |
| | 60≤ | 761 | 24.7 | 412 | 22.8 | 349 | 27.4 |
| Sex | Male | 12 | 0.4 | 7 | 0.4 | 5 | 0.4 |
| | Female | 3,068 | 99.6 | 1,800 | 99.6 | 1,268 | 99.6 |
| Payer type | NHI | 2,961 | 96.1 | 1,732 | 95.9 | 1,229 | 96.5 |
| | Medicaid | 119 | 3.9 | 75 | 4.2 | 44 | 3.5 |
| Preoperative period painkiller | No painkiller | 2,745 | 89.1 | 1,581 | 87.5 | 1,164 | 91.4 |
| | Step 1 painkiller | 214 | 7.0 | 150 | 8.3 | 64 | 5.0 |
| | Step 2 painkiller | 55 | 1.8 | 38 | 2.1 | 17 | 1.3 |
| | Step 3 painkiller | 66 | 2.1 | 38 | 2.1 | 28 | 2.2 |

ALND, axillary lymph node dissection; SLNB, sentinel lymph node biopsy; NHI, national health insurance

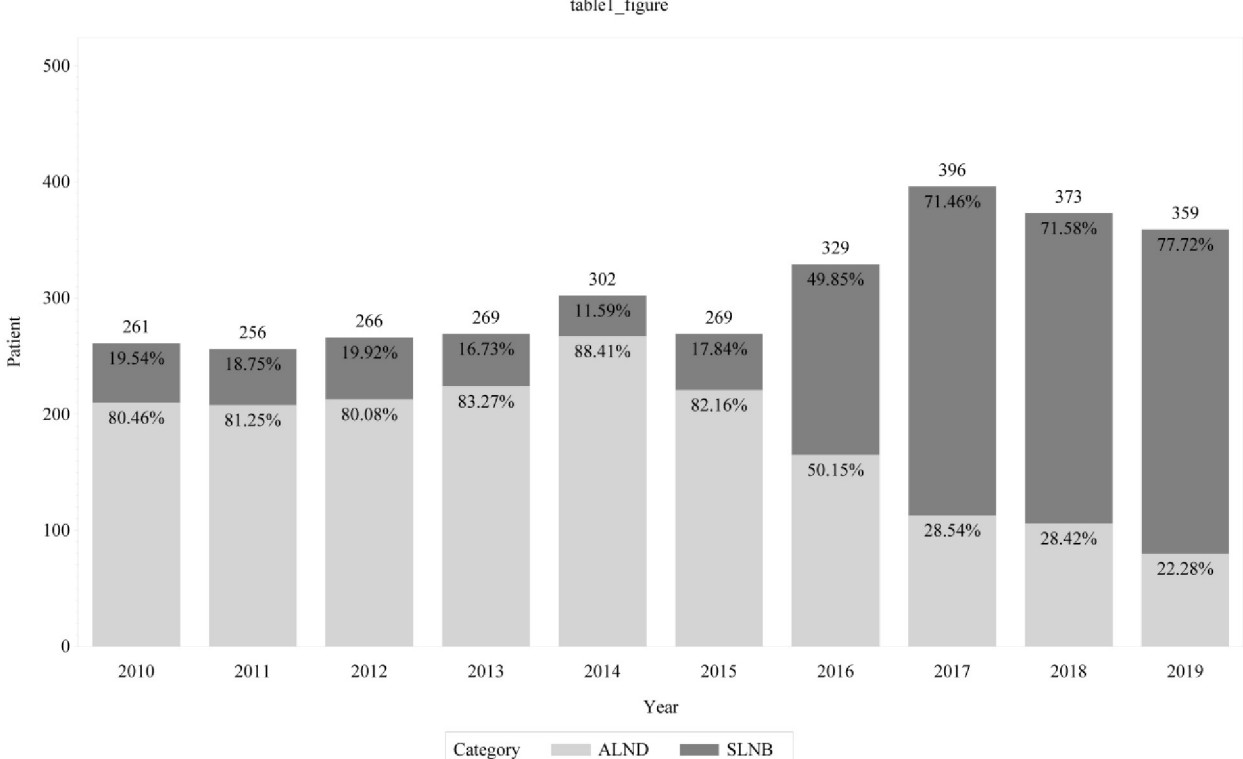

**Fig 3. Trends in the types of lymph node surgery from 2010 to 2019.** ALND: axillary lymph node dissection; SLNB: sentinel lymph node biopsy.

(3.77%) in the SLNB group. The percentage of patients prescribed postoperative pain medications was higher in the ALND group than in the SLNB group. In contrast, there was no significant difference between the two groups in terms of the percentage of painkiller prescriptions on the index date (non-opioid analgesics: 90.81% vs. 90.34%, tramadol: 34.92% vs. 36.21%, opioids: 94.63% vs. 94.58%). Overall, the percentages of preoperative/index date/postoperative

**Table 2. Preoperative drug utilization for pain management.**

| Drug category | ALND (n = 1,807) | | | | | | SLNB (n = 1,273) | | | | | |
|---|---|---|---|---|---|---|---|---|---|---|---|---|
| | Preoperative period | | Index date | | Postoperative period | | Preoperative period | | Index date | | Postoperative period | |
| | No. of patients | Percent | No. of patients | Percent | No. of patients | Percent | No. of patients | Percent | No. of patients | Percent | No. of patients | Percent |
| Non-opioid analgesics | 175 | 9.7% | 1,641 | 90.8% | 362 | 20.0% | 75 | 5.9% | 1,150 | 90.3% | 222 | 17.4% |
| Tramadol | 46 | 2.5% | 465 | 25.7% | 122 | 6.8% | 23 | 1.8% | 262 | 20.6% | 70 | 5.5% |
| Opioids | 38 | 2.1% | 1,710 | 94.6% | 91 | 5.0% | 28 | 2.2% | 1,204 | 94.6% | 48 | 3.8% |
| Anesthetics (N01AH*) | - | - | 1,640 | 90.8% | 9 | 0.5% | 1 | 0.1% | 1,168 | 91.8% | 4 | 0.3% |
| Analgesics (N02A*) | 38 | 2.1% | 631 | 34.9% | 84 | 4.6% | 27 | 2.1% | 461 | 36.2% | 45 | 3.5% |
| Anticancer | 253 | 14.0% | 186 | 10.3% | 1,074 | 59.4% | 112 | 8.8% | 77 | 6.0% | 756 | 59.4% |

ALND, axillary lymph node dissection; SLNB, sentinel lymph node biopsy

*Anatomical Therapeutic Chemical classification system

**Table 3. Association between postoperative opioid prescription and types of breast cancer surgery.**

| Surgery | Model | Opioids# including tramadol | | | Opioids# | | |
|---|---|---|---|---|---|---|---|
| | | Odds ratio | (95% CI) | P-value | Odds ratio | (95% CI) | P-value |
| SLNB (ref. ALND) | Model 1* | 0.727 | (0.546–0.970) | 0.030 | 0.764 | (0.504–1.158) | 0.205 |
| | Model 2† | 0.737 | 0.553–0.981 | 0.037 | 0.771 | (0.510–1.165) | 0.217 |
| | Model 3‡ | 0.718 | (0.538–0.959) | 0.025 | 0.761 | (0.503–1.151) | 0.196 |

*Model 1 adjusted for surgery year

†Model 2 adjusted for age, payer type, and surgery year

‡Model 3 adjusted for age, payer type, surgery year, preoperative period painkiller use, and preoperative period anticancer drug use

#Opioid category includes both anesthetic (N01AH*) and analgesic (N02A*) opioids as per the Anatomical Therapeutic Chemical classification system.

anticancer drug prescriptions were higher in the ALND group than in the SLNB group. The data for the prescription of drugs other than painkillers and anticancer drugs are shown in S2 Table.

All logistic regression models showed that the prescription of opioids, including tramadol, was significantly lower in the SLNB group than in the ALND group (Model 1: OR 0.727, 95% CI 0.546–0.970; Model 2: OR 0.737, 95% CI 0.553–0.981; Model 3: OR 0.718, 95% CI 0.538–0.959). In contrast, the prescription of opioids other than tramadol was lower in the SLNB group than in the ALND group; however, the difference was not statistically significant (Table 3).

## Discussion

In this study, we investigated the prescription status of postoperative pain medications for patients with breast cancer who underwent lymph node surgery in South Korea from 2010 to 2019 using NHI claim data from the HIRA and examined whether the type of surgery had an effect on the prescription of pain medications. The number of patients who underwent lymph node surgery showed an overall increasing trend over time; however, since 2014, the number of ALND cases has steadily decreased, while the number of SLNB cases has increased. The percentage of pain medication prescriptions on the index date was similar between the two patient groups, but for the prescription of postoperative pain medication and anticancer drugs, the prescription ratio was higher in the ALND patient group than in the SLNB patient group. In addition, the analysis based on logistic regression models to examine the effect of the lymph node surgery method on the prescription of postoperative pain medications demonstrated that the odds of opioid prescription were higher in the ALND group than in the SLNB group in most of the models used in the logistic regression analysis.

Recently, the rate of early detection of breast carcinoma has been increasing owing to the development of breast cancer screening and an increase in the screening rate. Accordingly, the surgical approach to breast cancer shows a trend in the surgical field of minimizing surgery-related morbidity and complications [25, 26]. In support of such trends, several clinical studies have reported that SLNB allows less extensive surgery without affecting oncological outcomes [8]. The publication of results from the American College of Surgeons Oncology Group (ACOSOG) Z0011 randomized clinical trial and dissemination of a regional guideline were associated with a considerable decrease in ALND rates [27]. Furthermore, SENTINA study showed SNLB to be a reliable diagnostic method with high detection rates before neoadjuvant chemotherapy [28], and SN FNAC study reported that the use of sentinel node biopsy after NAC can reduce unnecessary axillary node dissection by more than 30% [29]. The Z1071 study

published in 2013 shows a similar result, suggesting SLNB to a reliable alternative to ALND [30]. This study shows a corresponding trend of a continuous increase in SLNB rates since 2014. A prospective observational study using the Korean Breast Cancer Registry reported a steady decrease in the proportion of patients who underwent ALND between 2009 and 2018 (85.2% in 2009 to 47.5% in 2018) [31]. However, in the present study, a rapid change was observed in 2016. This study showed relatively different results, as it observed the trend of surgery occurrence in the entire population, whereas Cha et al. (2022) selected and tracked patients according to the same criteria as that of the Z0011 clinical trial. In comparison to the rapid decrease in ALND frequency observed in the United States and Europe following the publication of the ACOSOG Z0011 study results and guideline revisions, changes in surgical rates among Asian women with breast cancer with different baseline characteristics have been slower, owing to uncertainty regarding the effectiveness of the new strategy.

For patients with stage 1 and 2 breast cancer, postoperative pain management plays an essential role in minimizing the risk of chronic pain, collectively referred to as "persistent post-mastectomy pain" [32, 33]. To examine the status of pain management after lymph node surgery in patients with breast cancer, this study analyzed the percentage of prescriptions of pain medication according to surgery type for the preoperative period, index date, and postoperative period. The results showed that the percentage of patients who were prescribed severe pain medications was higher in the ALND group than in the SLNB group. According to a previous study, the prevalence of lymphedema, a postoperative complication, was high in the ALND patient group [34], and patients with lymphedema were prescribed more pain-related medications due to pain [35]. Taken together, recent trends in guidelines [36] and related studies suggest the need for careful attention to pain management for complications such as lymphedema, in addition to direct pain control resulting from surgery, owing to the significantly wider incision range and invasiveness of ALND compared with those of SLNB.

Anticancer therapy has also been reported to be a key factor in pain management in patients with breast cancer. Domestic and international guidelines recommend or approve neoadjuvant chemotherapy for patients with specific clinical conditions [37, 38]. However, according to a previous study by Schott et al., patients receiving neoadjuvant chemotherapy had a high risk of developing peripheral neuropathy, a representative side effect of chemotherapy, and other types of side effects, such as impaired cardiac function, nausea, and vomiting, which may affect surgical outcomes after chemotherapy [39, 40]. Lymph node metastasis is an important factor that influences the choice of surgical treatment. Because the presence of lymph node metastases could not be assessed from the insurance claim data used in this study, we controlled for the effect of lymph node metastases by adjusting for neoadjuvant chemotherapy. Model 3 showed that the trend of postoperative opioid prescriptions was consistent even after adjustment for neoadjuvant chemotherapy, indicating a significant difference in the trend of opioid prescriptions according to the surgical treatment method.

Opioids are an irreplaceable pharmacotherapy for postoperative pain management [41]. However, a recent study reported that the postoperative exposure of opioid-naïve patients to opioids resulted in approximately 10% of patients with cancer undergoing curative-intent surgery for chronic opioid users [42, 43]. Minimizing unnecessary exposure to opioids and implementing optimal pain management through an appropriate surgical method and postoperative pain management are crucial in managing long-term quality of life. South Korea has recently established standards for opioid prescription by presenting the opioid prescription guidelines for 2017 and 2021 [44]. However, no domestic guidelines have been explicitly set for the general use of pain medications, including opioids, for postoperative pain management in patients with breast cancer. The findings of this study are significant in that they demonstrate the status of drug prescriptions for postoperative pain management in this situation.

With the increasing survival rate of patients with breast cancer, improvement in their long-term quality of life and return to normal daily activities are considered the main treatment goals. Therefore, the clinicians' judgment to balance pain control and minimization of side effects, which have a trade-off relationship, plays a critical role. Healthcare professionals should be careful in selecting pain medications and adjusting doses according to the individual conditions and circumstances of patients and strive to improve the quality of life of patients by achieving maximal pain management with minimal side effects.

## Strengths and limitations

This study analyzed a database that adequately represents the national patient population of South Korea and examined the current status of lymph node surgery types and the prescription of pain medications for postoperative pain management in patients with breast cancer. Additionally, data from 2010 to 2019 were used to analyze relatively recent trends and patterns in healthcare service utilization. In particular, under the current circumstances, in which less-invasive surgical methods are preferred compared to clinical practices in the past, the association between the lymph node surgery method and the prescription of postoperative pain medication was investigated.

Nevertheless, this study has some limitations. First, as this was a cross-sectional study based on NHI claims, only the association, not the causal relationship, between the prescription of postoperative pain medication and the lymph node surgery method could be investigated, and it was not possible to examine the effect of disease severity (staging), clinical conditions of the patients, and extent of the disease. Furthermore, the cross-sectional nature of the database allowed only a limited number of patients who received surgery in the diagnosed year after the breast cancer diagnosis to be included in the study. Second, owing to the nature of HIRA claim data, the medical records of patients can only be viewed and assessed for up to 1 year; therefore, it was difficult to analyze or determine the average duration of drug prescription and changes in prescriptions for each patient in long-term. In addition, because the NPS data were obtained by a stratified sampling of approximately 2% of the Korean population, it may be difficult to generalize the findings of this study to the prescription patterns of all patients with breast cancer in South Korea. Third, the characteristics of prescription drugs were determined by the combination of the number of prescriptions and the number of prescription days, but specific details of prescriptions, such as the frequency of drug utilization, the total number of days of drug administration, and administered dose, could not be included in the analysis. Therefore, defined daily dose (DDD) and prescribed daily dose (PDD) of prescription opioids were not considered outcomes of this study. For more accurate and detailed information on drug utilization, an analysis based on DDD or PDD is required, and this needs to be considered in future studies.

## Conclusions

In this study, the NHI claim data provided by the HIRA were used to analyze the types of lymph node surgery patients with breast cancer underwent and the status of pain medication prescriptions for postoperative pain management. The findings of this study confirms the trend of less invasive interventions on lymph nodes, which led to lower odds of exposure to postoperative opioids. This reflects the importance of managing patients' long-term quality of life in the treatment of breast cancer and related pain control. In order to minimize postoperative pain and opioid use, close monitoring and observation of the patient's condition to select the appropriate surgical approach is considered an important step. In addition, careful

investigation of the long-term recurrence rate in patients with breast cancer is required, and follow-up studies are required for further examination and analysis of these issues.

## Supporting information

**S1 Table. Classification of medications.**
(DOCX)

**S2 Table. Perioperative drug utilization.**
(DOCX)

## Author Contributions

**Conceptualization:** Eun Sol Kang, Ye-Seul Lee.

**Data curation:** Yu-Cheol Lim.

**Formal analysis:** Yu-Cheol Lim.

**Funding acquisition:** Ye-Seul Lee.

**Investigation:** Eun Sol Kang.

**Methodology:** Yu-Cheol Lim.

**Project administration:** Yoon Jae Lee, In-Hyuk Ha, Ye-Seul Lee.

**Supervision:** In-Hyuk Ha, Ye-Seul Lee.

**Validation:** Bo-Hyoung Jang, Yoon Jae Lee.

**Writing – original draft:** Eun Sol Kang.

**Writing – review & editing:** Yu-Cheol Lim, Bo-Hyoung Jang, Yoon Jae Lee, In-Hyuk Ha, Ye-Seul Lee.

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
