## [Decision Letter · Decision Letter 0]

4 Oct 2023

PONE-D-23-19141Changes in lymph node surgery in breast cancer and preoperative drug prescription analysis for postoperative pain management: A retrospective, cross-sectional studyPLOS ONE

Dear Dr. Lee,

Thank you for submitting your manuscript to PLOS ONE. After careful consideration, we feel that it has merit but does not fully meet PLOS ONE’s publication criteria as it currently stands. Therefore, we invite you to submit a revised version of the manuscript that addresses the points raised during the review process.

We look forward to receiving your revised manuscript.

Kind regards,

Daniele Ugo Tari, M.D.

Academic Editor

PLOS ONE

Reviewers' comments:

Reviewer's Responses to Questions

**Comments to the Author**

1. Is the manuscript technically sound, and do the data support the conclusions?

Reviewer #1: Yes

2. Has the statistical analysis been performed appropriately and rigorously? 

Reviewer #1: Yes

3. Have the authors made all data underlying the findings in their manuscript fully available?

Reviewer #1: Yes

4. Is the manuscript presented in an intelligible fashion and written in standard English?

Reviewer #1: Yes

5. Review Comments to the Author

Reviewer #1: This is a retrospective analysis of patients treated for breast cancer with the aim of evaluating the use of opioid drugs through the type of axillary surgery (ALND or SLNB). I would like to congratulate the authors for the excellent work and suggest some modifications.

1- Abstract, data about the results obtained in this study must be included. We could not see the number of patients included and other important analyzes that should be in the summary.

2- Results: I did not understand why among 29,896 patients coded with breast cancer, more than 26,000 did not have surgery. This needs to be clarified better.

3- The type of surgery performed should be included in the demographic characteristics, as well as in the tables, as opioid prescriptions could be impacted due to the type of breast surgery performed. Multivariate analysis would also be important in relation to breast surgery. You should also evaluate the cases that underwent immediate reconstruction or not, as well as those that underwent oncoplastic techniques and the types of total reconstructions.

4- In the discussion, due to the results obtained during the study period, the data from the Z1071, SENTINA and SNFANC studies presented in the period 2012/2103 should be cited and discussed. Undoubtedly, these studies may have impacted this trend found in this analysis, even more than the Z11 results presented in 2010.

5- Conclusion: the main result of the study should be cited, which is the impact on opioid use.

6. PLOS authors have the option to publish the peer review history of their article (what does this mean?). If published, this will include your full peer review and any attached files.

Reviewer #1: **Yes: **Francisco Pimentel Cavalcante

---

## [Author Response · Author response to Decision Letter 0]

20 Dec 2023

Reviewer #1: 

This is a retrospective analysis of patients treated for breast cancer with the aim of evaluating the use of opioid drugs through the type of axillary surgery (ALND or SLNB). I would like to congratulate the authors for the excellent work and suggest some modifications.

- We thank the reviewer for his kind comments.

1- Abstract, data about the results obtained in this study must be included. We could not see the number of patients included and other important analyzes that should be in the summary.

- We thank the reviewer for the comment. We revised the abstract based on the reviewer’s comment as below:

This study aimed to investigate the changes in lymph node surgery types and prescription patterns of postoperative medications for pain management in patients with breast cancer using national health insurance claim data from South Korea. The study population comprised patients with at least one record of a principal diagnosis of breast cancer (ICD-10 code: C50) from the national health insurance claim database between 2010 and 2019. Patients who underwent mastectomy or lumpectomy only once were selected for the analysis. Patients who underwent axillary lymph node dissection (ALND) with mastectomy or lumpectomy on the day of surgery were included in the ALND group, whereas those who underwent sentinel lymph node biopsy (SLNB) were included in the SLNB group. Prescription records of opioids before, after and on the date of breast cancer surgery were collected and categorized according to the opioid type. Multivariate logistic regression modeling was used to compare postoperative opioid prescriptions. The proportion of those undergoing ALND among 3,080 patients decreased consistently after 2014, while the proportion undergoing SLNB increased. Although the rate of pain medication prescription on the day of surgery was similar between the two groups, the rate of prescription of postoperative pain medication and anticancer agents was lower in the SLNB group than in the ALND group. Logistic regression modeling showed that the SLNB group had lower odds of receiving opioids than did the ALND group (Odds ratio (OR) = 0.727, Confidence Interval (CI) = 0.546 - 0.970). A consistent trend was observed when the model was adjusted for neoadjuvant chemotherapy and the use of preoperative pain medications (OR = 0.718, CI = 0.538 - 0.959). To manage postoperative pain and prevent chronic pain with minimal side effects, sufficient discussion among clinicians, patients, and other healthcare professionals is imperative, along with adequate treatment planning.

2- Results: I did not understand why among 29,896 patients coded with breast cancer, more than 26,000 did not have surgery. This needs to be clarified better.

- We appreciate the reviewer’s comment and agree that the number of patients who received surgery in our data was significantly reduced in our study design. This is due to the limitation of our database obtained from National Agency, which was cross sectional and did not allow a follow-up during the consecutive year. In order to control possible heterogeneity in the cancer stage and severity of patients, we limited our study group to those who received surgery in the diagnosed year, and without any additional surgeries afterwards. This meant that patients who were diagnosed with breast cancer and received surgery in the following year were not analyzed.

We did consider adopting a simpler patient definition of those who received surgery in one year, but due to the cross-sectional nature of our database, this definition did not allow for adjustment of patients severity nor provided information on the patient’s prior history, i.e., cancer stage, neoadjuvant treatments, and previous surgeries, which would significantly confound the results. As a result, only 3,080 patients, of whom we were clearly able to analyze the preoperational and post-operational medications, were included in the study analysis.

Based on the reviewer’s comment, we revised the figure and Discussion as follows:

Fig 2

Discussion

First, as this was a cross-sectional study based on NHI claims, only the association, not the causal relationship, between the prescription of postoperative pain medication and the lymph node surgery method could be investigated, and it was not possible to examine the effect of disease severity (staging), clinical conditions of the patients, and extent of the disease. Furthermore, the cross-sectional nature of the database allowed only a limited number of patients who received surgery in the diagnosed year after the breast cancer diagnosis to be included in the study.

3- The type of surgery performed should be included in the demographic characteristics, as well as in the tables, as opioid prescriptions could be impacted due to the type of breast surgery performed. Multivariate analysis would also be important in relation to breast surgery. You should also evaluate the cases that underwent immediate reconstruction or not, as well as those that underwent oncoplastic techniques and the types of total reconstructions.

- We thank the reviewer for the comment. We agree that the types of breast surgery would influence subsequent opioid prescriptions. However, our database which covers the healthcare claims from 2010 to 2019 had limited information on the details of the types of breast surgery. For instance, the reimbursement of breast reconstruction was allowed in 2015 in Korea, and its operation was conducted as out-of-pocket before and during the initiation and amendment of the reimbursement. This led to a number of misinformation on detailed breast surgery codes. Therefore, we were advised by experts against including the details of breast surgery types in our analysis to avoid providing misinformation in our paper.

4- In the discussion, due to the results obtained during the study period, the data from the Z1071, SENTINA and SNFANC studies presented in the period 2012/2103 should be cited and discussed. Undoubtedly, these studies may have impacted this trend found in this analysis, even more than the Z11 results presented in 2010.

- We thank the reviewer for the comment. We revised the Discussion based on the reviewer’s comment as below:

In support of such trends, several clinical studies have reported that SLNB allows less extensive surgery without affecting oncological outcomes [8]. The publication of results from the American College of Surgeons Oncology Group (ACOSOG) Z0011 randomized clinical trial and dissemination of a regional guideline were associated with a considerable decrease in ALND rates [27]. Furthermore, SENTINA study showed SNLB to be a reliable diagnostic method with high detection rates before neoadjuvant chemotherapy [28], and SN FNAC study reported that the use of sentinel node biopsy after NAC can reduce unnecessary axillary node dissection by more than 30% [29]. The Z1071 study published in 2013 shows a similar result, suggesting SLNB to a reliable alternative to ALND [30]. This study shows a corresponding trend of a continuous increase in SLNB rates since 2014.

5- Conclusion: the main result of the study should be cited, which is the impact on opioid use.

- We thank the reviewer for the comment. We revised the Discussion based on the reviewer’s comment as below:

In this study, the NHI claim data provided by the HIRA were used to analyze the types of lymph node surgery patients with breast cancer underwent and the status of pain medication prescriptions for postoperative pain management. The findings of this study confirms the trend of less invasive interventions on lymph nodes, which led to lower odds of exposure to postoperative opioids. This reflects the importance of managing patients’ long-term quality of life in the treatment of breast cancer and related pain control. In order to minimize postoperative pain and opioid use, close monitoring and observation of the patient's condition to select the appropriate surgical approach is considered an important step. In addition, careful investigation of the long-term recurrence rate in patients with breast cancer is required, and follow-up studies are required for further examination and analysis of these issues.

---

## [Decision Letter · Decision Letter 1]

23 Jan 2024

Changes in lymph node surgery in breast cancer and preoperative drug prescription analysis for postoperative pain management: A retrospective, cross-sectional study

PONE-D-23-19141R1

Dear Dr. Lee,

We’re pleased to inform you that your manuscript has been judged scientifically suitable for publication and will be formally accepted for publication once it meets all outstanding technical requirements.

Kind regards,

Daniele Ugo Tari, M.D.

Academic Editor

PLOS ONE

Additional Editor Comments (optional):

Reviewers' comments:

Reviewer's Responses to Questions

**Comments to the Author**

1. If the authors have adequately addressed your comments raised in a previous round of review and you feel that this manuscript is now acceptable for publication, you may indicate that here to bypass the “Comments to the Author” section, enter your conflict of interest statement in the “Confidential to Editor” section, and submit your "Accept" recommendation.

Reviewer #1: All comments have been addressed

2. Is the manuscript technically sound, and do the data support the conclusions?

Reviewer #1: Yes

3. Has the statistical analysis been performed appropriately and rigorously? 

Reviewer #1: Yes

4. Have the authors made all data underlying the findings in their manuscript fully available?

Reviewer #1: Yes

5. Is the manuscript presented in an intelligible fashion and written in standard English?

Reviewer #1: Yes

6. Review Comments to the Author

Reviewer #1: (No Response)

7. PLOS authors have the option to publish the peer review history of their article (what does this mean?). If published, this will include your full peer review and any attached files.

Reviewer #1: **Yes: **Francisco Pimentel Cavalcante

---

## [Editor Report · Acceptance letter]

26 Mar 2024

PONE-D-23-19141R1 

PLOS ONE

Dear Dr. Lee, 

I'm pleased to inform you that your manuscript has been deemed suitable for publication in PLOS ONE. Congratulations! Your manuscript is now being handed over to our production team.

Kind regards, 

on behalf of

Dr. Daniele Ugo Tari 

Academic Editor

PLOS ONE